# Protocol for the development and validation of a patient-reported experience measure (PREM) for people with hearing loss: the PREM-HeLP

Helen Pryce,[1] Sian Karen Smith [ID],[1] Georgina Burns-O'Connell,[1] Rebecca Knibb [ID],[2] Rosemary Greenwood,[3] Rachel Shaw [ID],[2] Saira Hussain,[1] Jonathan Banks [ID],[4] Amanda Hall [ID],[1,5] Jean Straus,[6] Sian Noble[7]

[1]Audiology, School of Optometry, Aston University College of Health and Life Sciences, Birmingham, West Midlands, UK
[2]School of Psychology, Aston University, Birmingham, UK
[3]University Hospitals Bristol and Weston NHS Foundation Trust, Bristol, UK
[4]Social & Community Medicine, University of Bristol, Bristol, UK
[5]Children's Hearing Centre, University Hospitals Bristol NHS Foundation Trust, Bristol, UK
[6]NIHR ARC Northwest London, London, UK
[7]School of Social and Community Medicine, University of Bristol, Bristol, UK

**Correspondence to**
Dr Helen Pryce;
h.pryce-cazalet@aston.ac.uk

## ABSTRACT

**Introduction** Hearing loss is a common chronic health condition and adversely affects communication and social function resulting in loneliness, social isolation and depression. We know little about the patient experience of living with hearing loss and their views on the quality of the audiology service. In this study, we will develop and validate the first patient-reported experience measure (PREM) to understand patients' experiences of living with hearing loss and their healthcare interactions with audiology services.

**Methods and analysis** We will develop the PREM in three phases: (1) development of PREM prototype (items/statements) derived from previous qualitative work and narrative review, (2) cognitive interview testing of the PREM prototype using a 'think aloud' technique to examine the acceptability and comprehensibility of the tool and refine accordingly and (3) psychometric testing of the modified PREM with 300 participants to assess the reliability and validity of the tool using Rasch analyses with sequential item reduction. Eligible participants will be young people and adults aged 16 years and over who have hearing loss. Participants will be recruited from three clinical sites located in England (Bath, Bristol) and Scotland (Tayside) and non-clinical settings (eg, lip-reading classes, residential care settings, national charity links, social media).

**Ethics and dissemination** The study was approved by the West of Scotland Research Ethics Service (approval date: 6 May 2022; ref: 22/WS/0057) and the Health Research Authority and Health and Care Research Wales (HCRW) Approval (approval date: 14 June 2022; IRAS project ID: 308816). Findings will be shared with our patient and public involvement groups, academics, audiology communities and services and local commissioners via publications and presentations. The PREM will be made available to clinicians and researchers without charge.

## BACKGROUND

Worldwide, hearing loss affects approximately 430 million adults and is the second largest chronic health condition contributing to the global burden of disease.[1] By 2050, hearing loss is estimated to affect around 700 million (1 in 10) of the global population. In the UK, around 12 million (1 in 5) adults have some measure of hearing loss, which is set to rise to 14.2 million by 2035.[2] Of concern is that up to 40% of UK adults aged over 50 years live with hearing loss, rising to 70% of adults over the age of 70 years.[3] The risk of hearing loss in older adults doubles among lower socioeconomic populations.[4]

Evidence consistently shows that hearing loss has broad and significant implications. Difficulties in communication and listening situations result in people feeling frustrated and ashamed about their hearing loss, and consequently withdrawing, minimising or avoiding social situations.[5 6] Hearing loss drives social isolation, social disengagement and feelings of loneliness.[7] Recent work has highlighted the array of negative emotions (frustration, anger, resentment, distress, embarrassment) and forms of fatigue (effort- and emotion-driven fatigue) experienced by people with hearing loss on

## STRENGTHS AND LIMITATIONS OF THIS STUDY

⇒ This PREM work will be based on concepts identified through comprehensive qualitative investigation and narrative review to identify common features of lived experience of hearing loss.
⇒ The PREM will be developed using codesign principles that bring together patient and staff experience and relevant stakeholder input.
⇒ Rasch analysis will be employed with subscales to identify whether subgroups, based on the age, gender or ethnicity of the responders, found different items more important than others.
⇒ Rasch analysis will investigate whether the dimensionality of the tool can be ignored in favour of a much shorter form that might have better uptake when used within a clinical setting.



a daily basis.[8 9] The societal stigma surrounding hearing loss is well documented and reinforces feelings of shame and threatens social identity.[10] With fewer opportunities to interact with others, hearing loss leads to poorer mental health.[11–13] Hearing loss is associated with other chronic diseases, including arthritis, cancer, cardiovascular risk factors, diabetes, stroke, visual impairment and mobility problems.[14] There is also growing evidence that hearing loss is associated with a greater risk of developing dementia, although the pathways connecting the two are ambiguous.[15–17]

Hearing loss cannot be cured and the illness burden is significant. In the UK, the commissioning of audiology services varies depending on which part of the country you live. The introduction of 'Any Qualified Provider' (AQP) services means that large corporations and optical chains have been awarded NHS contracts to provide audiology services in England.[18] AQP service providers in England are primarily funded around provision of hearing aids with limited time to learn about the patient's lived experience.[19 20] For mild to moderate hearing loss, hearing aids have shown to ameliorate hearing function and quality of life through improving communication and relationships.[21–23] The visibility of hearing aids as an indicator of hearing loss, and the negative connotations associated with ageing and hearing loss, continue to carry much stigma.[24–27] Acclimatising to hearing aids can be arduous as people have to learn how to adapt to amplified sounds.[28]

The 'burden of treatment' framework is a useful way to consider the many demands placed on patients to manage their chronic conditions.[29 30] In the context of hearing loss, we can distinguish between illness work (living with hearing loss) and treatment work (accessing and managing hearing aids). Examples of illness work include recognising symptoms and signs of hearing loss in relation to internalised illness representations and managing the negative emotions (distress, shame) associated with hearing loss. In terms of treatment work, patients are almost entirely responsible for managing their hearing aids. This requires practical and technical work, including regularly wearing the aid, making sure the device is at the correct volume and replacing the fine tubing. New features of hearing aids have required patients to come up to speed with technology, for example, learning how to use Bluetooth to pair devices.[31] Hearing aid non-use (including people who struggle to manage hearing aids) is often attributed to the hearing aid user's ability or motivation rather than a reflection on the burdensome work involved in managing them.[24] The illness and treatment burden increase as we age due to the accumulation of health conditions.[14 30]

There is growing support for the use of self-report, validated questionnaires, namely patient-reported outcome measures (PROMs), patient-reported experience measures (PREMs) and patient satisfaction measures in measuring the quality of care received to inform service changes.[32] Although complementary, there are important differences between PROMs, PREMs and patient satisfaction measures and what they aim to achieve.[33 34] PROMs are used to evaluate the effectiveness of treatment or intervention from the patient perspective.[35] In audiology, PROMs have been developed to assess the impact of hearing loss or perceived benefit of hearing aids on quality of life.[36] By contrast, PREMs focus on the patients' viewpoint of the overall experience of the care they have received to gain insight into the patient centredness of services.[37] This might include patients' perceptions of how clearly health professionals communicated and how adequate they found the care coordination.[38] PREMs and patient satisfaction measures are also not synonymous. Satisfaction relates to the patient's expectations of the care and whether their expectations were met during healthcare encounters.[39] Unlike PREMs, patient satisfaction instruments do not capture the emotions and challenges experienced when living with a chronic condition.[40]

Up until recently most research has predominately focused on the design and application of PROMs, with relatively less attention on PREM development and implementation.[41] However, PREM development is gaining momentum for several health conditions (eg, diabetes, chronic kidney disease, inflammatory bowel disease, rheumatoid arthritis).[37 42 43] In the UK, the Care Quality Commission and National Institute for Health and Care Excellence have produced guidelines emphasising the importance of patient experience,[44] while the Office for Health Improvement and Disparities endorses the use of patient outcome measures.[32] Patient experience not only relates to the experience of care, but also how people live with and feel about their condition. This is reflected in PREM work for chronic obstructive pulmonary disease (COPD) which measures the emotional impact of living with COPD alongside experiences of care.[45–47]

In audiology, it is important that service delivery is grounded in patient experience to develop a holistic understanding of the daily illness and treatment burdens of hearing loss. This will help to identify gaps in audiology care and improve service provision. For example, how clearly clinicians explain diagnostic procedures, how much distress patients experience resulting from explanations of processes. Furthermore, providing routine information on residual burdens arising from hearing loss (eg, managing social withdrawal in difficult listening environments) could be supported by existing provision in audiology care, and identification of specific needs could facilitate tailored care, for example, referral to hearing therapy, lip-reading classes, assistive listening devices, etc. Although audiology departments may have their own measures to understand patient experience, there are currently no validated instruments that measure how people feel about their hearing loss and the work involved in coping with illness and treatment burdens. The current study will develop, pilot and validate a novel PREM to capture the experience of living with hearing loss and audiology care, from

the patient perspective. This proposed work forms part of a larger study—the Hearing Loss and Patient-Reported Experience study (HeLP study)—the first study of its kind to build an understanding of living with hearing loss and treatment burden across the life course. While not the focus of the current study, the first stage of the HeLP study—qualitative interviews to explore the lived experience of hearing loss[48]—will inform the development of the PREM by identifying key themes to guide the items. The objectives will be to:

1. Develop a PREM prototype informed by qualitative interviews and literature review.
2. Test the PREM prototype using a think aloud approach.
3. Test the reliability and validity of the PREM.

## METHODS

This study will comprise three phases, with each phase examining one of the objectives described above. Figure 1 gives an overview.

### Phase 1: development of PREM prototype

Our qualitative work exploring the lived experience of coping with hearing loss and navigating audiology services at different life stages will be used to generate preliminary items for the PREM.[48] A systematic review will collate international evidence to gain a broad understanding of the key topics underpinning the lived experience of hearing loss. Combined, the qualitative work and review will inform the development of a conceptual model to explain how coping with hearing loss is experienced

---

**Phase 1: PREM prototype development**

Development of preliminary set of items

*Item generation* informed by qualitative interviews with people with hearing loss, narrative review exploring the lived experience, and input from PPI groups

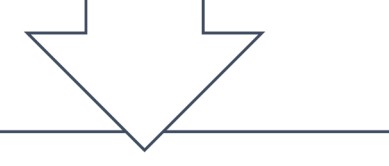

**Phase 2: Cognitive testing of the PREM prototype**

Think-aloud testing

Cognitive interviews with 10 patients and 10 clinicians recruited from clinical sites (Bath, Bristol and Scotland)

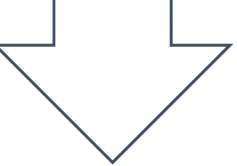

**Phase 3: Psychometric testing of the PREM**

Reliability and validty testing

300+ service users, recruited from clinical and non-clinical routes (including clinical research networks)

**Figure 1** Overview of study design and patient recruitment strategies for PREM development, piloting and validation. PREM, patient-reported experience measure; PPI, patient and public involvement.

and negotiated alongside different personal, social and cultural contexts. It will provide a theoretical explanation of the trade-offs that people make between using hearing aids and managing without and seeking clinical help or not. This in turn will guide the initial list of scale items covering key themes. Additionally, a rapid review of PREM literature will be undertaken to ensure the PREM is informed by current and robust evidence and recommendations. To ensure the items are clear and resonate with patients, we will phrase items using participants' words and language. We will also draw on the burden of treatment theory to consider the different aspects of the experience required to manage hearing loss, including both 'illness work' (responding to and coping with hearing loss) and 'treatment work' (the efforts required to access and participate in care).[29]

PREM development will be an iterative activity with researchers drafting a prototype PREM, and patient and public involvement (PPI) and service users providing feedback. The researchers will actively engage with marginalised groups to ensure the PREM is reviewed by a wide range of potential service users, including adults with English as a second language, adults with learning disabilities, adults with dementia, residents in care homes and adults who have both chosen, and not chosen, to engage with audiology services. We will also advertise to members of lip-reading classes, and use PPI engagement with typically marginalised community groups (eg, South Asian women's exercise classes) to reach participants who may not be attending clinical sites.

The initial list of PREM items will be reviewed, refined and agreed by the research team and PPI groups (Public and Patient Involvement). Any disagreement on the inclusion of an item will be discussed further in relation to the data generated from Work Package 1 until agreement is reached. Each item will be assessed for clarity, readability, necessity and overlap with other items. Items will be concise and designed to be understood and interpreted in the same way by all patients. The PREM prototype will be assessed for readability using reading age software (readble.io) and language will be adjusted to meet requirements for a reading age of 7 years (below UK average reading age). The PREM will also be worded so that it can be used as carer-proxy scales for patients who are not able to complete the PREM themselves. This continual engagement and revision process will refine the prototype PREM in preparation for cognitive testing.

### Phase 2: cognitive testing of the PREM prototype

We will use a think aloud approach to pilot the PREM prototype and identify whether the items are acceptable and cover issues that are important and relevant to people with hearing loss.[48] Guided by sample sizes used in previous think aloud work,[49] we will aim to recruit and invite 10 people with hearing loss via the three clinical sites located in England (Bath, Bristol) and Scotland (Tayside) and non-clinical groups (eg, lip-reading classes, mosques, residential care, social media). Eligible

participants will be young people and adults aged 16 years and over who have hearing loss and the capacity to give informed consent. Participants who took part in our previous qualitative work will be invited to take part in this next stage of the research and provide feedback on the PREM to refine the scale. We will also recruit five to eight clinicians from our clinical sites to obtain their views on the tool.

The think aloud interviews will ask participants to respond to each item while verbalising their thoughts. The researcher will probe further to explore participants' understanding of each item and how it could be scored. This will allow us to ascertain the clarity of both concepts and language and revise any items that are not understood as intended for the target population. In addition, we will ask whether any important questions are missing. We will seek views on the overall length of the PREM, the response mechanisms, general structure and suggestions for improvement. This process will enable us to establish content validity by ensuring that all items are relevant and reflect hearing loss comprehensively. Four researchers (HP, SKS, GB-O'C, SH) will separately conduct and analyse the think aloud interviews to ensure a range of participants from different demographics and age groups are included. All think aloud interviews will be audio-recorded, transcribed verbatim and analysed using thematic analysis techniques to identify and group common responses to the PREM questions inductively.[50] For example, it provides a way of grouping descriptions of completing the prototype, linking common features of the meaning statements into broader patterns (eg, use of narrative to describe specific scenarios that related to the items) or emotional references to experiences that are triggered by questions. It enables researchers to identify patterns in response to items on clinical care versus items on lived experience, and how the thought process altered when having to imagine a scenario based on recent experiences or past care experiences. Common responses will be grouped into themes, for example, where the phrasing of a question seems to slow down the response or where the individual seeks clarification before responding. Where responses appear affected by ambiguity or inconsistency between participants' responses, discussion with the wider research team and PPIE (Patient and Public Involvement and Engagement) consultation will consider alternative wording. Agreement on the final themes will be reached through discussion between coauthors. Agreed themes will be used to inform revisions to the PREM items, for example, if wording of an item is ambiguous or difficult to measure on a prescribed response scale.

### Phase 3: psychometric testing of the PREM reliability and validity

Following required modifications (identified in phase 2), the reliability and validity of the PREM will be evaluated to ensure the tool generates consistent results under similar circumstances and measures what it intends to measure.[37] The revised PREM will be distributed to

people with hearing loss who will have consented to take part in the HeLP research study. As mentioned above, participants will be recruited from the audiology departments, clinical research networks, social/residential care settings, lip-reading classes, national charity links and social media adverts. The PREM will be administered via a secure online server (Qualtrics), alongside validated scales, for content validity analysis. To reach social care settings, further completion of the questionnaires will be conducted in person with researchers visiting residential care settings.

Guidelines suggest that a minimum of 300 participants are optimal for scale development and we will therefore aim to recruit this number or recruit a minimum of 10 individuals for each item in the prototype scale, whichever is the largest sample size.[51] Evidence from Work Package 1 of the HeLP study (a detailed large-scale qualitative description of experience) will enable us to identify the specific constructs that will apply to the PREM validation. As previous research into the lived experience of hearing loss suggests isolation and depression are associated with hearing loss, construct validity will be assessed by examining correlations between our hearing loss PREM and measures of social isolation, decisional conflict and the UCLA Loneliness Scale.[52] We will use a generic quality of life scale (the EQ-5D 5L) to consider the relationship between PREM items and quality of life responses,[53] and a measure of health literacy such as Chew's health literacy screening questions,[54] but use our PPI to support the choice of which tools to use.

Participants will be asked to complete the PREM 2 weeks after initial completion to assess retest reliability. Item reduction will include identifying items with poor levels of completion and those without discriminant properties where virtually all participants have recorded the same response category. Internal consistency of these domains will be assessed using Cronbach's alpha. Internal structural validity of the scale(s) will be assessed using exploratory factor analysis (non-orthogonal rotational method) to investigate or confirm the number of domains using factor selection with Eigen values above 1.

To improve the consistency and reliability, several split samples will be extracted using different random splits and the process repeated at 10 times. Rasch analysis will be employed with these subscales to identify whether subgroups based on the age, gender or ethnicity of the responders found different items more important than others. It will also be used to further investigate whether there is an opportunity to reduce the number of items based on relative importance of the item from the qualitative data, lack of fit to the Rasch model, redundancy detected earlier and evidence of a response bias that would adversely impact its use in a diverse population. A further Rasch analysis will be attempted to investigate whether the dimensionality of the tool can be ignored in favour of a much shorter form that might have better uptake when used within a clinical setting.

Pearson's bivariate correlations will be conducted between scale scores to assess construct validity (how well the scale measures the lived experience of hearing loss). The sample size will allow investigation of correlations as low as 0.2 to be statistically significant of the scale. Between-subjects t-tests and Pearson's correlations will assess discriminative validity of the scale by comparing across demographic and hearing loss characteristics, and health resource use.

### Study steering committee group
The study steering group comprises eight independent members who will oversee the HeLP project and advise the study research team, funder and sponsor of the research. Specifically, it includes one international collaborator, two experts by lived experience, three clinicians working in audiology in different parts of the UK, one social scientist and chaired by a health commissioner with a particular interest in patient centred care. Steering group meetings will be held every 6 months to monitor project progress and delivery milestones.

### Study research team
The study team comprises three clinician-researchers and eight researchers with complementary skills and expertise in clinical experience, methodology and topic. The study will be led and coordinated by the Chief Investigator (HP), an academic and hearing therapist with 31 years of clinical experience in audiological rehabilitation. Our team includes a health psychologist with expertise in scale development and testing (RK) who will oversee the statistical methods together with a medical statistician (RG) with expertise in research design. Our researchers include a sociologist with interest in hearing (GB-O'C), two health psychologists (RS and SKS) with experience of conducting applied qualitative and mixed methods research, two academic-clinical scientists (SH and AH) working in audiology and a qualitative researcher (JB) with interest implementation science. Our expert by experience (JS) and PPI lead will provide PPI insights. Our PPI researcher works in marginalised community settings (including residential care homes) to provide access to perspectives of both help seekers and non-help seekers with hearing loss. Our health economist (SN) provides oversight and commentary of the impact on health service use.

### Patient and public involvement
Two PPI leads (a researcher with PPI responsibility and a public member) will manage and organise the PPI activities, working closing with the lead researcher and the research team. Our PPI leads have experience with people who have complex communication needs and will speak to marginalised individuals in settings that suit them, for example, residential homes. Likewise, our younger PPI group have requested online contact. Guidance and steering will be sought from our PPI members at every stage of the research process. PPI activities will

include reviewing patient information sheets and consent forms, advising on recruitment and strategy, planning think aloud interview approach and checking coding and analysis procedures. In relation to the PREM, the PPI members will advise on PREM format, content and comprehensibility. Fieldnote diaries of PPI input, activities, engagement and outcomes will be kept to record the role PPI members play in shaping the research and PREM.

Our PPI approach considers populations who are at greater risk of hearing loss, including people from South Asian communities (Bangladeshi, Indian and Pakistani), adults with learning disabilities and older adults living in residential care. The following groups have been consulted[1]: South Asian community groups from local religious centres to broaden access to those from Muslim backgrounds,[2] individuals living in residential care homes (with whom we have existing connections),[3] Aston PPI group (a local group who experience hearing loss, including younger student members aged 18–40 years who meet virtually)[4] and Bath PPI group (a long-standing group of older adults who advise and support on healthcare delivery and research). This provides a mix of people who volunteer to give feedback through established PPI groups and individuals giving feedback via interview or email to our PPI community leads directly through contact with community groups.

## Ethics approval

This study was reviewed and approved by the West of Scotland Research Ethics Service (approval date: 6 May 2022; ref: 22/WS/0057) and the Health Research Authority and Health and Care Research Wales Approval (approval date: 14 June 2022; IRAS project ID: 308816). Study participation is voluntary and participants can withdraw at any time. All data collected from participants will be anonymous and kept confidential and de-identified by allocating participants in each study with a unique ID. Participating in the research may enable participants to reflect on their experience of hearing loss and develop new insights which could be enabling. At the same time, if a participant becomes distressed during an interview or while completing the PREM, he/she will have the choice of pausing or stopping the interview or survey.

All research data will be stored securely on the university password protected server and only accessed by researchers directly associated with the study. For the cognitive think aloud interviews, written informed consent will be obtained from participants prior to interview and hard copies stored in a secure filing cabinet at the university only accessible to study staff. Participants who choose to take part in an online interview will be asked to post their consent form to the university using a reply-paid envelope. The researcher will record verbal consent at the start of the interviews. For the psychometric testing phase (online survey), participants will be provided with an electronic consent form to read and sign online.

Encrypted digital recorders or Microsoft Teams recording software will be used to audio-record interviews. Recordings will be deleted once transcribed and checked for accuracy. Data will be stored in line with the Data Protection Act 2018 and General Data Protection Regulation standards, and study documents (paper and electronic) will be kept in a secure location for 5 years at Aston University. Any personal data collected (eg, contact details) will be held separate from research data on the secure Aston University Box file.

## Dissemination of the study findings and PREM measure

To disseminate our findings, we will seek out opportunities to deliver presentations at relevant academic and non-scientific conferences and meetings. We will publish articles in appropriate peer-reviewed journals read by our target audience and professional magazines (eg, British Academy of Audiology (BAA) Audacity) to ensure findings are circulated among audiology healthcare professionals and academics. The PPI groups, audiology services and local commissioners will also share findings, and social media (eg, Twitter) will be used to spread awareness of our work to our network of followers.

Once the PREM is developed, it will be made available free of charge (under a Creative Commons Licence) to support its implementation within the clinical sites: Bath, Bristol and Tayside. The tool will be complemented by targeted implementation resources for clinicians, for example, instructional training videos, leaflets and software compatible with existing patient management systems. The implementation phase will be written up as a separate protocol publication. Ultimately, the team will seek to raise awareness and promote the roll-out of the PREM into wider audiology practices across the UK. For example, we will disseminate the tool to colleagues and stakeholders via conferences (British Society of Audiology (BSA), BAA, 'Hearing Across the Lifecourse'). We will also approach BSA special interest groups who produce professional guidance on practice. Our project website will include downloadable pdf versions.

**Acknowledgements** The study is sponsored by University Hospitals Bristol and Weston NHS Foundation Trust, and supported by Aston University Applied Audiology research group. We would also like to extend our gratitude and thanks to the PPI members for their ongoing steering and support for the HeLP study.

**Contributors** HP is the lead researcher leading the protocol development, ethical approval, data collection, data analysis and dissemination. SKS wrote the first draft of the protocol and final manuscript. GB-OC, RS, SH, JB, AH, RK, RG, JS and SN have provided feedback and support in the development of the protocol and study documents including the ethics application and interview schedule. All authors read, contributed to, edited and agreed the final manuscript.

**Funding** This study is supported by an NIHR HSDR grant (Funding stream REF NIHR 131597). The sponsor for this study is the University Hospitals Bristol and Weston NHS Foundation Trust. Aston University (Applied Audiology research group, College of Health and Life Sciences) is the supporting institution. Dr Jonathan Banks is partly funded by National Institute for Health and Care Research Applied Research Collaboration West (NIHR ARC West) and NIHR Health and Social Care Delivery HS&DR (REF NIHR 131597).

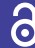

**Competing interests** None declared.

**Patient and public involvement** Patients and/or the public were involved in the design, or conduct, or reporting, or dissemination plans of this research. Refer to the Methods section for further details.

**Patient consent for publication** Not applicable.

**Provenance and peer review** Not commissioned; externally peer reviewed. This protocol was reviewed by the South West Research Design Service Advisory Group, members of the Research Design Service, Aston University Research Governance team, PPI groups and patient experts.

**ORCID iDs**
Sian Karen Smith http://orcid.org/0000-0002-9541-2221
Rebecca Knibb http://orcid.org/0000-0001-5561-0904
Rachel Shaw http://orcid.org/0000-0002-0438-7666
Jonathan Banks http://orcid.org/0000-0002-3889-6098
Amanda Hall http://orcid.org/0000-0001-8520-6005

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
