## [Reviewer comments · BMJ Open]

ARTICLE DETAILS

TITLE (PROVISIONAL)	Protocol for the development and validation of a patient reported experience measure (PREM) for people with hearing loss: the PREM- HeLP
AUTHORS	Pryce, Helen; Smith, Sian; Burns-O'Connell, Georgina; Knibb, Rebecca; Greenwood, Rosemary; Shaw, Rachel; Hussain, Saira; Banks, Jonathan; Hall, Amanda; Straus, Jean; Noble, Sian

VERSION 1 – REVIEW

REVIEWER	Jack Holman University of Nottingham, Hearing Sciences - Scottish Section
REVIEW RETURNED	01-Aug-2023

GENERAL COMMENTS	Thank you for the opportunity the review this study protocol. I believe that the proposed study is of great value and brings together some pressing issues in hearing healthcare today. There are a few changes I believe are necessary and questions to be answered before the protocol should be published. I would like a bit more explanation on the use of the scale. There are nice definitions of PREMs as being used to inform service change and as distinct from PROMs. However, I don't quite see how the experience of life with hearing loss aspect of the PREM (as opposed to the experience of healthcare provision) would be used to inform service change. Perhaps an example or some other explanations would benefit this. I also found myself asking whether the PREM is intended to be used in targeted settings, on multiple occasions, as routine practice, for audit, will there be benefit from use alongside well known and well used PROMs etc. Pg 5 In 10: The sentence starting "Hearing aid non-use (including people who struggle" needs to be reworked. I think I know what is intended but currently it is confusing as it suggests in part that people struggling to manage hearing aids is not attributed to the work involved in managing hearing aids. Pg 5 In 16: spelling/grammar "accumulation of by multiple" Acronyms need to be sorted throughout. For example: PPI should be defined. Define WP1. Health and Care Research Wales is abbreviated, but the other two ethics boards are not. They should only be abbreviated if this is useful later on in the manuscript. BSA and BAA are not defined. Pg 7 In 16: I am interested to know how you will identify and interact
---

	with individuals with hearing loss who have not interacted with audiology services. It is of course not impossible, but usually a hearing loss is diagnosed by audiology. Pg 8 In 6: Could the authors provide more information on the analysis plan. Thematic analysis is not a pre-boxed one-size-fits-all methodology so more detail would be necessary. Pg 8 In 26: spelling/grammar “participants are optimal development”...”therefore” Pg 8 In 29:It is not entirely clear what is planned regarding assessment of construct validity. Research has identified many psychosocial factors that are related to hearing loss, so there should be justification for choosing these two (isolation and depression). The next phrase stating that “we will use EQ-5D-5L and HLQ” does not make it clear what they will be used for. It should state if the mentioned scales are certain to be included and that only additional tools will be decided by the PPI group. Pg 11: To start with I am very happy to see a well-developed and progressive approach to PPI work. Please explain more about the composition of the PPI groups. Are there four that already exist? It seems like some are well defined whereas others are not, or at least maybe don't consist of a defined list of names. Will there be a requirement for responses from each potential subgroup (e.g. white/Asian, young/middle aged/old)? It would just be good to get a bit more clarity on this. Pg 12: Are there/will there be data transfer agreements in place for authors based in different institutions?
--	--

VERSION 1 – AUTHOR RESPONSE

Comment	Our response
Thank you for the opportunity the review this study protocol. I believe that the proposed study is of great value and brings together some pressing issues in hearing healthcare today. There are a few changes I believe are necessary and questions to be answered before the protocol should be published. I would like a bit more explanation on the use of the scale. There are nice definitions of PREMs as being used to inform service change and as distinct from PROMs. However, I don't quite see how the experience of life with hearing loss aspect of the PREM (as opposed to the experience of healthcare provision) would be used to inform service change. Perhaps an example or some other explanations would benefit this. I also found myself asking whether the PREM is intended to be used in targeted settings, on multiple occasions, as routine	Thank you for your positive comments. We have added more explanation on the intended use of the PREM on page 6. For example, how clearly clinicians explain diagnostic procedures, how much distress or anticipation patients experience resulting from explanations of processes. Furthermore, providing routine information on residual burdens arising from hearing loss (e.g. managing social withdrawal in difficult listening environments) could be supported by existing provision in audiology care, and identification of specific needs could facilitate better tailored care e.g. referral to Hearing Therapy; lipreading classes; assistive listening devices etc.

practice, for audit, will there be benefit from use alongside well known and well used PROMs etc.	
Pg 5 In 10: The sentence starting “Hearing aid non-use (including people who struggle” needs to be reworked. I think I know what is intended but currently it is confusing as it suggests in part that people struggling to manage hearing aids is not attributed to the work involved in managing hearing aids.	We have edited the second part of the clause to simplify the sentence. Hearing aid non-use (including people who struggle to manage hearing aids) is often attributed to the hearing aid user’s ability or motivation rather than a reflection on the burdensome work involved in managing them (24).
Pg 5 In 16: spelling/grammar “accumulation of by multiple”	Thanks for picking this up – edited.
Acronyms need to be sorted throughout. For example: PPI should be defined Define WP1. Health and Care Research Wales is abbreviated, but the other two ethics boards are not. They should only be abbreviated if this is useful later on in the manuscript. BSA and BAA are not defined.	PPI and WP1/BSA/BAA are now written in full Removed abbreviation of ethics board
Pg 7 In 16: I am interested to know how you will identify and interact with individuals with hearing loss who have not interacted with audiology services. It is of course not impossible, but usually a hearing loss is diagnosed by audiology.	We have added text to explain direct advertisement. We have both included clinical sites to recruit individuals who are engaging with audiology services and community links to those who are not currently actively engaging with audiology services. The following text has been added on page 7: By directly advertising to care home residents; members of lip-reading classes and using PPI engagement with typically marginalised community groups (e.g. South Asian Women’s exercise classes) we will reach participants who are not currently attending clinical sites.

Pg 8 In 6: Could the authors provide more information on the analysis plan. Thematic analysis is not a pre-boxed one-size-fits-all methodology so more detail would be necessary.	We have provided specific detail on the use of thematic analysis in this context on page 8. All think aloud interviews will be audio-recorded, transcribed verbatim and analysed using thematic analysis techniques (50). Thematic analysis is a process to inductively gather data through 'think aloud' completion of the PREM prototype. For example, it provides a way of grouping descriptions of completing the prototype; linking common features of the meaning statements into broader patterns e.g. participants using narrative to describe specific scenarios that related to an item, or items eliciting an emotional response. It enables researchers to identify patterns in responses to items on clinical care versus items on daily lived experience e.g. how did the thought process alter when having to imagine a scenario and respond based on recent experiences or past care experiences. Common responses will be grouped into themes in responses. Agreement on the final themes will be reached through discussion between co-authors. Agreed themes will be used to inform revisions to the PREM items e.g., if wording of an item is ambiguous or difficult to measure on a prescribed response scale.
Pg 8 In 26: spelling/grammar "participants are optimal development"... "therefore"	Thank you - corrected
Pg 8 In 29: It is not entirely clear what is planned regarding assessment of construct validity. Research has identified many psychosocial factors that are related to hearing loss, so there should be justification for choosing these two (isolation and depression). The next phrase stating that "we will use EQ-5D-5L and HLQ" does not make it clear what they will be used for. It should state if the mentioned scales are certain to be included and that only additional tools will be decided by the PPI group.	We have re worded this paragraph accordingly. Until we had some findings from our first work package (a large-scale qualitative study on the lived experience) it was difficult to anticipate what concepts would need to be included in the PREM and therefore what validation tools would be best. We have edited this with better understanding (at this point in time) of what we will be planning. (52). Evidence from work package 1 of the HeLP study which provided a detailed large scale qualitative description of experience will enable us to identify the specific constructs that will apply to the PREM validation. As previous research into the lived experience of hearing loss suggests isolation and depression are associated with hearing loss, construct validity will be assessed

	by examining correlations between our hearing loss PREM and measures of social isolation; decisional conflict, and the UCLA Loneliness scale (54). We will use a generic quality of life scale (the EQ-5D 5L) to consider the relationship between PREM items and quality of life responses (55), and a measure of health literacy such as the self-reported health literacy questions (56), but use our PPI to support the choice of which tools to use.
Pg 11: To start with I am very happy to see a well-developed and progressive approach to PPI work. Please explain more about the composition of the PPI groups. Are there four that already exist? It seems like some are well defined whereas others are not, or at least maybe don't consist of a defined list of names. Will there be a requirement for responses from each potential subgroup (e.g. white/Asian, young/middle aged/old)? It would just be good to get a bit more clarity on this.	That's correct. We were mindful that existing PPI groups do not necessarily represent the views of more marginalised communities in audiology and therefore went out to community groups and care homes to seek additional views. We have added a sentence (page 12) to reflect this: This provides a mix of people who volunteer to give feedback through established PPI groups, and individuals giving feedback via interview or email to our PPI community leads directly through contact with community groups.
Pg 12: Are there/will there be data transfer agreements in place for authors based in different institutions?	Contracts are in place between the contributors in different institutions. Only Aston researchers are gathering, analysing and storing data.

VERSION 2 – REVIEW

REVIEWER	Jack Holman University of Nottingham, Hearing Sciences - Scottish Section
REVIEW RETURNED	02-Nov-2023

GENERAL COMMENTS	Thank you to the authors for the re-submitted work. I am happy with most of the responses and alterations following my comments, but I would like a little bit more information on one point before publishing. The authors have given a nice overview of the benefits of using thematic analysis; however I would prefer a little bit more information on the practicalities of the chosen thematic analysis process. The authors mention that information will be inductively gathered which is important, but it would be helpful to state other details such as how many researchers will be involved in initial coding and the iterative steps involved in code refining and theme development (as it's unlikely that all co-authors will be involved at every stage). Perhaps alluding to the iterative process would be of benefit. The process to be undertaken sounds like reflexive thematic analysis to me, so if so, it would be helpful to add this also.
--

	I look forward to reading about the PREM when it is complete.
--	---

VERSION 2 – AUTHOR RESPONSE

Comment	Our response
Please revise the ‘Strengths and limitations of this study’ section of your manuscript (after the abstract). This section should contain up to five short bullet points, no longer than one sentence each, that relate specifically to the methods. The novelty, aims, results or expected impact of the study should not be summarised here.	Thank you for this clarification – we have edited our points accordingly to focus on proposed methods.
The authors have given a nice overview of the benefits of using thematic analysis; however I would prefer a little bit more information on the practicalities of the chosen thematic analysis process. The authors mention that information will be inductively gathered which is important, but it would be helpful to state other details such as how many researchers will be involved in initial coding and the iterative steps involved in code refining and theme development (as it’s unlikely that all co-authors will be involved at every stage). Perhaps alluding to the iterative process would be of benefit.	Thank you for this point. We have added further detail on P.8 to illustrate the process and confirmed that 4 researchers will engage in ‘think aloud’ interviews and analysis of responses in an effort to broaden the potential range of responses e.g., by accessing a range of postcode districts across the England and Scotland, age and help-seeking status of respondents.
The process to be undertaken sounds like reflexive thematic analysis to me, so if so, it would be helpful to add this also.	We would prefer to describe this as using thematic analysis methods e.g. we are collating and synthesising common features of responses but given that this is not a purely inductive phase of work (albeit based on substantive grounded theory work in package 1 – see Pryce, H., Smith, S.K., Burns-O’Connell, G., Shaw, R., Hussain, S., Banks, J., Hall, A., Knibb, R., Greenwood, R. and Straus, J., 2023. Protocol for a qualitative study exploring the lived experience of hearing loss and patient reported experience in the UK: the HeLP study. BMJ open, 13(6), p.e069363.) In this case participants are engaging in ‘think aloud’ interviews with a specific set of questions and therefore is unlikely to lead to full reflexive thematic analysis as interviews are not using sufficient open ended interview strategies.